# Sensible heat has significantly affected the global hydrological cycle over the historical period

G. Myhre [1], B.H. Samset [1], Ø. Hodnebrog [1], T. Andrews [2], O. Boucher [3], G. Faluvegi [4,5], D. Fläschner[6], P.M. Forster [7], M. Kasoar [8], V. Kharin[9], A. Kirkevåg [10], J.-F. Lamarque [11], D. Olivié[10], T.B. Richardson [7], D. Shawki[8], D. Shindell [12], K.P. Shine [13], C.W. Stjern [1], T. Takemura [14] & A. Voulgarakis [8]

Globally, latent heating associated with a change in precipitation is balanced by changes to atmospheric radiative cooling and sensible heat fluxes. Both components can be altered by climate forcing mechanisms and through climate feedbacks, but the impacts of climate forcing and feedbacks on sensible heat fluxes have received much less attention. Here we show, using a range of climate modelling results, that changes in sensible heat are the dominant contributor to the present global-mean precipitation change since preindustrial time, because the radiative impact of forcings and feedbacks approximately compensate. The model results show a dissimilar influence on sensible heat and precipitation from various drivers of climate change. Due to its strong atmospheric absorption, black carbon is found to influence the sensible heat very differently compared to other aerosols and greenhouse gases. Our results indicate that this is likely caused by differences in the impact on the lower tropospheric stability.

[1] CICERO Center for International Climate Research – Oslo, 0318 Oslo, Norway. [2] Met Office Hadley Centre, Devon, EX1 3PB, United Kingdom. [3] Institut Pierre-Simon Laplace, CNRS/Sorbonne Université, 75252 Paris, Cedex 05, France. [4] NASA Goddard Institute for Space Studies, New York, NY 10025, USA. [5] Center for Climate Systems Research, Columbia University, New York, NY 10027, USA. [6] Max-Planck-Institut für Meteorologie, 20146 Hamburg, Germany. [7] University of Leeds, Leeds,, LS2 9JT, United Kingdom. [8] Department of Physics, Imperial College London, London,, SW7 2AZ, United Kingdom. [9] Canadian Centre for Climate Modelling and Analysis, V8P 5C2 Victoria, BC,, Canada. [10] Norwegian Meteorological Institute, 0313 Oslo, Norway. [11] NCAR/UCAR, 80305 Boulder, CO, USA. [12] Duke University, 27708 Durham, NC, USA. [13] University of Reading, Reading, RG6 6BB, United Kingdom. [14] Kyushu University, 816-8580 Kasuga, Japan. Correspondence and requests for materials should be addressed to G.M. (email: gunnar.myhre@cicero.oslo.no)

Multiple lines of evidence indicate widespread changes to the global water cycle since 1950[1]. Over land, however, the overall change in precipitation is observed to be small on average[2], but with a highly inhomogeneous regional pattern. While measured trends in precipitation are still uncertain[3–5] they indicate an increase in precipitation at high latitudes, and in parts of the tropics, while the subtropics and certain mid-latitude regions have seen reductions[1,6]. Over oceans, a similar pattern of precipitation change can be deduced from observed changes in salinity[1]. Modelled precipitation change also broadly agrees with the pattern from observations[2,7], albeit with significant inter-model variability in magnitude and the exact geographical distribution[7]. Increases in heavy precipitation are also found in both models and observations, in line with theoretical expectation of the response of the global water cycle to anthropogenic climate change[8–10].

Precipitation is strongly linked to the energy budget in the atmosphere, as the surface latent heat flux is a direct heat source for the atmosphere when water vapour condenses[11–13]. The total energy associated with evaporation and condensation is large, amounting to around one quarter of the incoming solar radiation[14]. The Earth's global energy budget, is, in turn, nearly in balance at the top of the atmosphere, at the surface and through the atmosphere[14]. To maintain this global-mean balance, any change in precipitation ($dP$) must be balanced by changes in atmospheric radiative cooling ($dQ$) and changes in the surface sensible heat flux ($dSH$):

$$L\,\mathrm{d}P = \mathrm{d}Q - \mathrm{d}SH, \qquad (1)$$

here $L$ is the latent heat of vaporization.

Changes in atmospheric radiative cooling via shortwave or longwave radiation can occur through the direct influence of a climate forcing mechanism, such as increased insolation or $CO_2$ concentration, or as a climate feedback through changes in temperature, water vapour or clouds. Enhanced atmospheric radiative cooling is known to be driven mainly by increased tropospheric temperatures associated with the surface warming[15,16]. Sensible heat, the transfer of heat from the surface to the atmosphere without any phase change, is dependent on the temperature difference between the surface and the overlying air, on turbulence and on convection. It is currently among the most uncertain factors in the present-day global energy budget[14], and its response to climate change is even less well understood. General circulation models indicate a small reduction in the upward SH from the surface[13–15,17] in response to $CO_2$-induced warming. Therefore, changes in precipitation are often assumed to be mainly balanced by radiative cooling, assuming small changes in SH. Here we show a substantial role of SH in contributing to current precipitation changes in state-of-the-art climate models. We further show that compensating factors influence the radiative cooling term so that the importance of the generally weaker magnitude of the SH term varies over time.

Here we combine climate model results from the Coupled Model Intercomparison Project Phase 5 (CMIP5)[18] and the Precipitation Driver and Response Model Intercomparison Project (PDRMIP)[19,20] to investigate the role of changes in radiative cooling and sensible heat that are associated with historical and future precipitation changes (see further description in the Methods section). Through simulations that individually perturb $CO_2$, methane ($CH_4$), black carbon (BC), sulphate ($SO_4$) and solar forcing, we also explore how different climate forcing mechanisms influence the sensible heat. In PDRMIP, dedicated simulations to understand change occurring on fast and slow timescales have been performed (Methods section).

## Results

**Importance of sensible heat changes in global climate model simulations.** Andrews et al.[21] showed that the precipitation change occurring on a fast timescale scales strongly with changes in atmospheric radiative cooling due to the direct effect of the forcing mechanism, while the slow, surface temperature driven precipitation change scales with top-of-atmosphere radiative forcing (because the surface temperature change is itself driven by the top-of-atmosphere forcing). This has recently been supported by a multi-model intercomparison (PDRMIP) study[20]. Based on the same ten PDRMIP models, Fig. 1 shows the relation between changes in precipitation, radiative cooling and sensible heat, i.e. the left and right hand sides of Eq. 1, when perturbing five different drivers of climate change, for the fast, slow and total response. For all drivers and timescales, we find that the atmospheric energy budget is closed, as anticipated from Eq. 1. In addition, by comparing the points that include radiative cooling changes, but not sensible heat changes, Fig. 1 illustrates the significant role of sensible heat for the atmospheric energy budget, and in particular on the fast timescale (Fig. 1a) for some of the drivers of climate change. In the remainder of this paper, we describe fast changes to precipitation, which depend on the drivers of climate change and slow (calculated from the difference between total and fast) changes, which are caused by surface (and consequently atmospheric) temperature changes.

Having established that present climate models maintain atmospheric energy balance for all drivers, we can study the interplay between sensible heat and radiative cooling for precipitation change in past and future simulations. Historical and future changes in precipitation and sensible heat for a selection of CMIP5 models (see Supplementary Table 6 for list of models) are shown in Fig. 2. Future changes are shown for the two most extreme emission scenarios of the representative concentration pathways (RCP) used within CMIP5 (RCP2.6 and RCP8.5)[22]. Historical precipitation changes from the CMIP5 climate models are weak, with reductions during periods of strong volcanic forcing (such as the period between 1960 and 2000). Towards the end of the CMIP5 historical period, in 2005 and towards the start of the future simulations with the RCP scenarios, the global precipitation notably increases. This trend continues over the 21st century, but the magnitude differs between the scenarios. Note, however, that while the model differences in precipitation are substantial at the end of the 20th century, the predicted magnitude of the increase over the 21st century is quite similar among the models, since standard deviations of $LdP$ in both cases are close to $1.0\ \mathrm{W\,m^{-2}}$ (see Supplementary Fig. 1 for individual CMIP5 models). The change in sensible heat as given in Eq. 1 is shown as a positive contribution to precipitation change, and thus with the sign convention of a reduction in sensible heat. Over the historical period, the sensible heat at the surface is gradually reduced in the CMIP5 models, and hence contributes to an increase in precipitation. For the 7-year period from 2010 to 2016, the precipitation and sensible heat changes since 1850 in the multi-model mean are equal, and amount to a global mean of $0.80\ \mathrm{W\,m^{-2}}$ (or, equivalently, a precipitation increase of 10 mm/year). Note that the multi-model mean increase in precipitation is equal to the reduction in surface sensible heat for the period from 2010 to 2016; however, this is not the case for all individual CMIP5 models (see Supplementary Fig. 1).

The important implication of the changes in precipitation and sensible heat being about the same at present conditions relative to 1850 are that, according to Eq. 1, the change in global atmospheric radiative cooling must be negligible at this time. For future precipitation changes, on the other hand, the contribution from enhanced radiative cooling dominates, with sensible heat contributing only modestly in the RCP8.5 scenario. In the RCP2.6

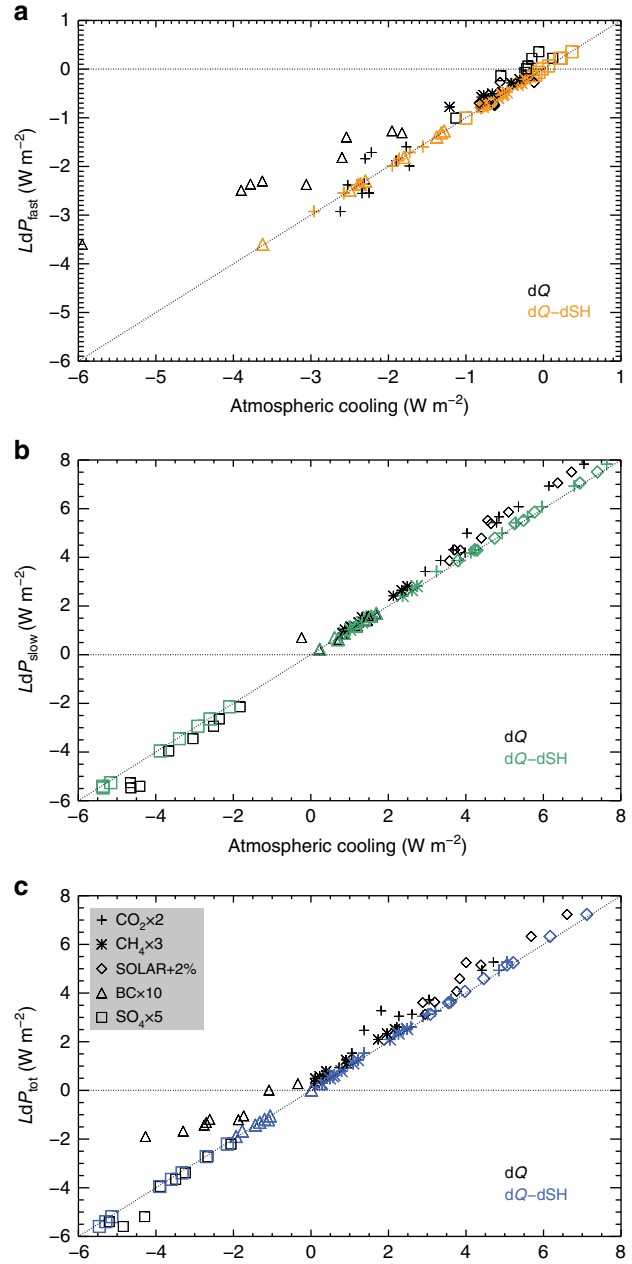

**Fig. 1** Modelled relation between changes in global-mean precipitation and atmospheric energy balance. Fast (**a**), slow (**b**) and total (**c**) precipitation change, as a function of changes in atmospheric cooling. Coloured symbols include changes in sensible heat according to Eq. 1, whereas black symbols exclude the sensible heat changes. The dotted line is the 1:1 relation of Eq. 1. Results are shown for five different drivers of climate change as described in the Methods section and their symbols are given in **c**. Results from ten PDRMIP climate models are included (Methods section)

and RCP4.5 scenario, the sensible heat changes are nearly constant over the 21st century (see Fig. 2 and Supplementary Fig. 2). Note, however, that for the 2010–2016 period, the relative standard deviation (RSD; defined as the standard deviation among the CMIP5 models divided by the multi-model mean) is close to four times larger for the predicted change in precipitation than for sensible heat. Hence, while sensible heat has a large contribution to the current precipitation change in the CMIP5 models, the inter-model diversity in the radiative cooling is the

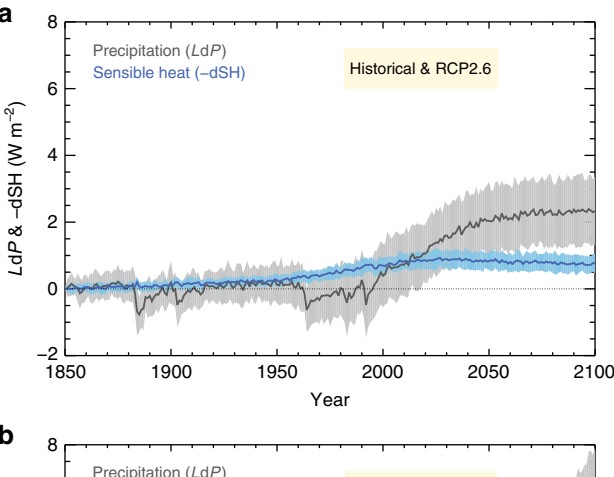

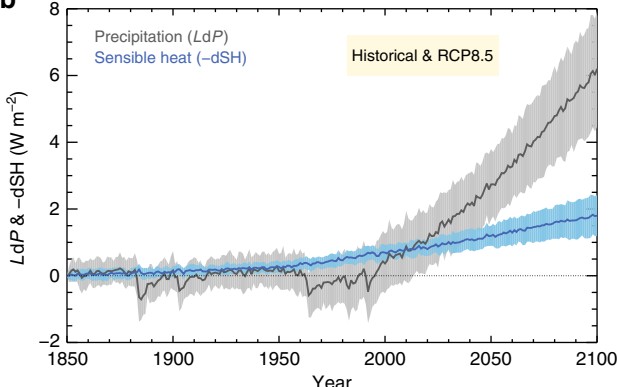

**Fig. 2** Global mean historical and future precipitation and sensible heat changes, from a selection of CMIP5 models. Future simulations are based on the RCP2.6 (**a**) and RCP8.5 (**b**) scenarios. The solid line shows the multi-model mean of the CMIP5 models, and ±1 standard deviation across the model sample is shown as a band. The individual models are shown in Supplementary Fig. 1. Sensible heat is given as a reduction in the upward surface flux and is thus a contribution to increased precipitation according to Eq. 1

main cause of the inter-model variation in precipitation. The RSD for sensible heat at the end of the 21st century increases by 20% relative to the period 2010–2016, whereas for precipitation it reduces by a factor of 3–4 for the RCPs. The difference in the historical surface temperature change among the CMIP5 models is a major cause of the difference in the simulated change in precipitation (see Supplementary Fig. 3b). On the other hand, the difference in BC climate impact among the CMIP5 models seems to cause only a small difference in the precipitation change (Supplementary Fig. 3a). While BC has been shown to affect precipitation in substantially different ways than other drivers of climate change[20,21,23], it is difficult to discern any impact on simulated historical precipitation change in CMIP5 due to differences in BC abundances (Supplementary Fig. 3). Figure 3 shows the geographical distribution of changes to sensible heat in CMIP5, over the historical period. The reduction in sensible heat (contributing to increase in global-mean precipitation) is large over ocean and over certain land areas such as South East Asia and parts of Africa and Europe. An increase in sensible heat (leading to global-mean precipitation reduction) is simulated in the CMIP5 model mean at high latitudes and central and South America.

**Sensible heat changes for various climate drivers**. In Figs. 4 and 5, we study the mechanisms behind the strong importance of the sensible heat for current precipitation changes in the CMIP5 ensemble, and their weaker importance for future changes.

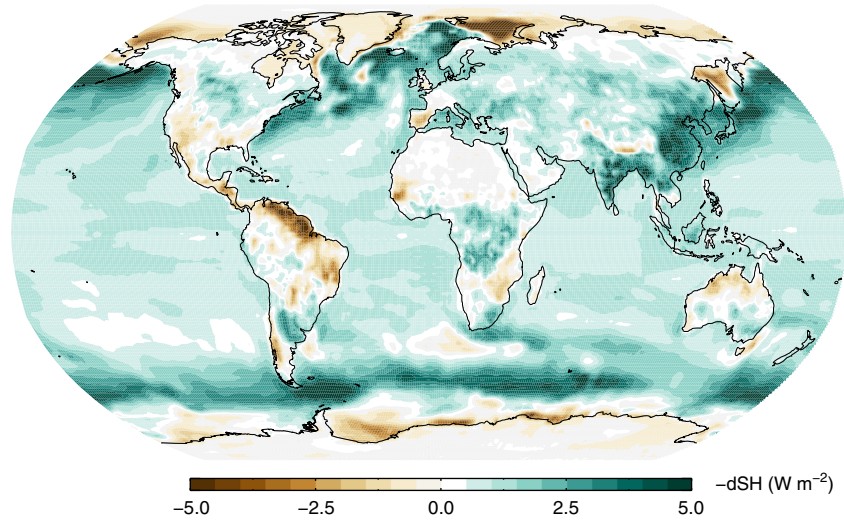

**Fig. 3** Geographical distribution of multi-model CMIP5 mean change in sensible heat between 2001–2005 and 1861–1865. Sensible heat is given as a reduction in the upward surface flux and is thus a contribution to increased global-mean precipitation according to Eq. 1

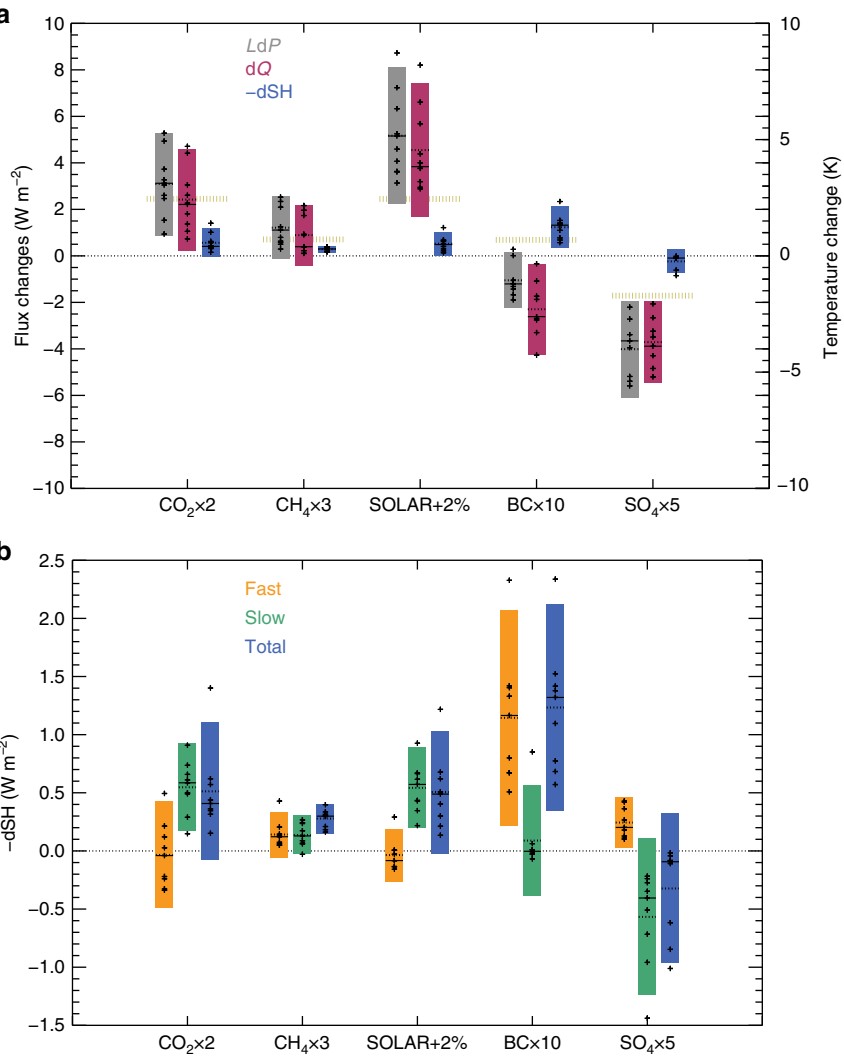

**Fig. 4** Influence of climate drivers on sensible heat and radiative cooling. **a** Total changes (fast plus slow) in sensible heat, radiative cooling, and precipitation (*LdP*) for five drivers of climate change from PDRMIP, **b** fast, slow and total change in sensible heat. Changes in sensible heat shown as –dSH from Eq. 1. The boxes show the 5–95% range based on the PDRMIP models. Dots show individual model values, solid line the median and dotted line the mean. Yellow dotted lines in **a** are the global multi-model mean surface temperature change. For clarity, the results from HadGEM3 for the $SO_4 \times 5$ case are not shown in **a** due to the much stronger forcing than in the other PDRMIP models, with *LdP* of around $-16\,W\,m^{-2}$

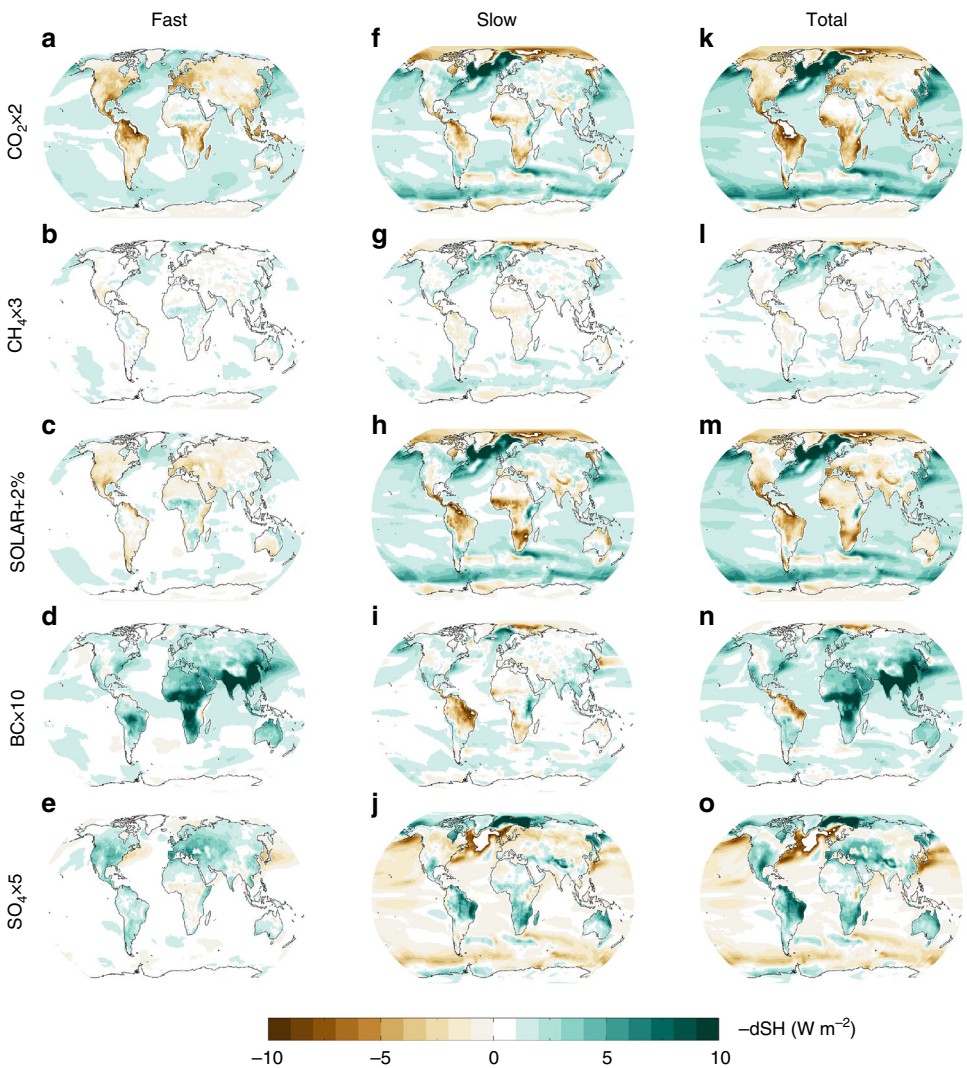

**Fig. 5** Fast, slow and total changes in surface sensible heat flux for five drivers of climate change. The fast changes are shown for $CO_2 \times 2$, $CH_4 \times 3$, SOLAR + 2%, BC × 10, $SO_4 \times 5$, in **a–e**, slow changes in **f–j** and total changes in **k–o**. Note that the figure shows reduced sensible heat from the surface to the atmosphere and the contribution of sensible heat to increase in global-mean precipitation and thus –dSH according to Eq. 1. Results are shown for the multi-model mean of the PDRMIP models

In Fig. 4a the sensible heat and the radiative cooling contribution to the precipitation change are shown for idealized changes in different drivers of climate change. The global, annual mean values normalized by the near-equilibrium surface temperature change are shown in Supplementary Fig. 4. The sensible heat changes are split into fast and slow changes (Fig. 4b). Normalizing the perturbations to show an increase in surface temperature (i.e. changing the sign of the negative sulphate forcing), all the drivers and models show an overall reduction in sensible heat flux from the surface to the atmosphere and thus a contribution to precipitation increases. BC differs substantially from the other drivers, with a large decrease in the sensible heat occurring on a fast timescale (Fig. 4b). This change in sensible heat offsets a substantial part of the reduction in the atmospheric radiative cooling (i.e. atmospheric absorption) due to BC, and thus dampens the reduction in precipitation. For drivers of climate change other than BC, the radiative cooling term dominates over the sensible heat term, consistent with Fig. 1. The RSD is, overall, relatively similar for sensible heat and the fast atmospheric radiative heating, whereas radiative cooling caused by surface and tropospheric temperature changes shows a lower uncertainty except for BC (see Supplementary Table 5).

The geographical distribution of changes in sensible heat (Fig. 5) shows that the overall reduction (and thus the contribution to increase in precipitation) mainly arises from changes over the ocean. For a doubling of $CO_2$ there is a general increase in sensible heat over land, both for fast and slow changes. The reduction over ocean dominates for the slow changes, whereas the changes over land and ocean almost cancel each other in the ensemble-mean for the fast changes[24]. The slow changes in sensible heat are similar for the other climate drivers when normalized by surface temperature changes, with some small exceptions for BC over part of South America and Africa (see Supplementary Fig. 5). The fast change in sensible heat is distinctly different for BC compared to the other climate drivers, with strong reductions over land. This decrease in sensible heat over land for BC, occurring on a fast timescale, explains the global-mean difference in sensible heat found for BC compared to the other climate drivers in Fig. 4. The changes in sensible heat in the CMIP5 models, shown in Fig. 3, have a pattern over ocean similar to the total sensible heat change in the PDRMIP $CO_2 \times 2$ simulation. Over land, however, the CMIP5 pattern more closely resembles that of the aerosol simulations in PDRMIP. This is particularly visible over South East Asia and parts of Europe and Africa.

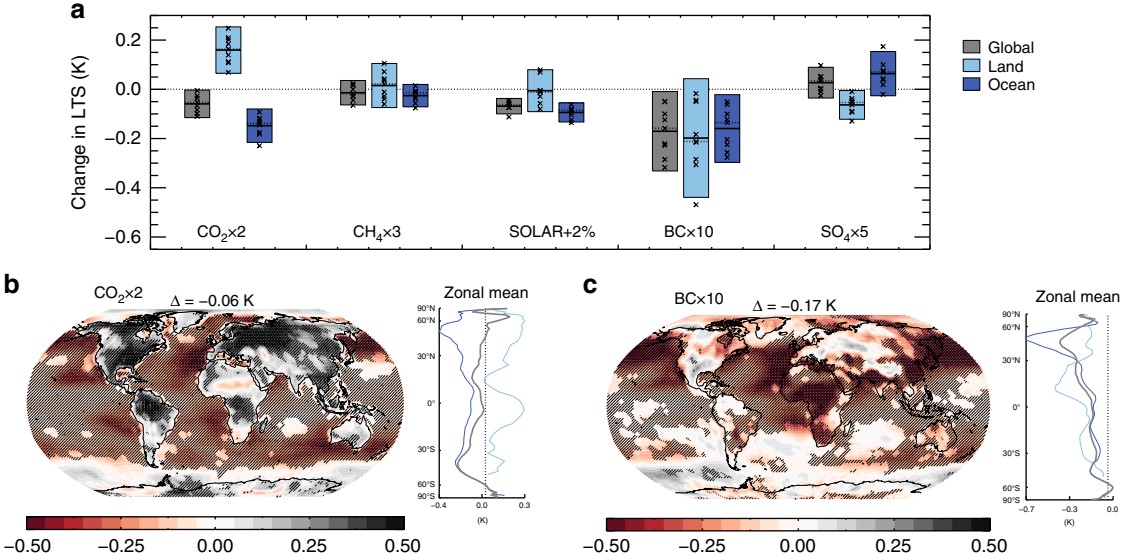

**Fig. 6** PDRMIP model-median changes in lower tropospheric stability (LTS). The LTS is defined as the temperature in the vertical layer corresponding to 1000 hPa minus the temperature in the 780 hPa-layer, based on years 6–15 in fixed-SST simulations. **a** Shows global, land only and ocean only changes in LTS, where individual crosses show individual model results, and the solid black line gives the model median. **b**, **c** show model median maps of LTS changes for $CO_2 \times 2$ and $BC \times 10$, respectively, including zonal mean changes for the global (grey), land (light blue) and ocean (blue). A reduction in the LTS (smaller temperature difference) indicates a more stable atmosphere in the lower troposphere. Surface pressure can at some locations be lower than 1000 hPa, especially at high altitudes over land. The 1000 hPa level represent the lowest model layers. Hatching is included where 75% or more of the models agree on the sign of the change

**Changes in components relevant for sensible heat**. Observations indicate that BC can reduce turbulence in the atmospheric boundary layer[25], which is likely to influence the surface sensible heat flux. These observations are in line with modelling of the stability in the lower troposphere within PDRMIP[26]. A major factor causing the reduced sensible heat from the surface, and particular the fast response over land, is the strong influence on the lower tropospheric stability (LTS) from BC, compared with the other climate drivers investigated in PDRMIP[26]. This is illustrated in Fig. 6, where changes in the sensible heat resemble many of the changes in LTS. LTS is here defined as the vertical temperature difference between the model levels corresponding to 1000 and 780 hPa. We find LTS changes consistent with the 1000–780 hPa response for other choices of model levels in the lower atmosphere as shown in Supplementary Fig. 6. When the LTS change is negative, this reflects an increase in the lower atmospheric stability. Figure 6a shows the global, land and ocean average LTS changes in the fixed-sea surface temperature (SST) experiments. The LTS global-mean change is stronger for BC than the other PDRMIP drivers, with a model-median change of −0.17 K. Over land, BC is the climate driver in the PDRMIP simulations with the strongest reduction in LTS with a median change of −0.21 K. $CO_2$ tends to reduce stability over land (positive change to LTS and hence increase in SH), and increase it over oceans (and hence a decrease in SH). Over ocean, the LTS change for $CO_2$ is found to be equally strong as for the BC perturbation. A hemispheric asymmetry is also clearly visible in Fig. 6c, with stronger LTS reductions over the Northern Hemisphere, where BC has been increased the most. LTS changes are given also for the other PDRMIP drivers in Supplementary Figure 7. Supplementary Fig. 8 shows the surface (1000 hPa) and 780 hPa temperature changes in the $CO_2 \times 2$ and $BC \times 10$ fixed-SST experiments. The land surface temperature changes at mid to high northern latitudes are somewhat larger in the $CO_2 \times 2$ experiment than in the $BC \times 10$ experiment, but at 780 hPa over regions with high BC abundance the temperature changes are larger in the $BC \times 10$ than in the $CO_2 \times 2$ experiment, explaining

the LTS change patterns. The tendency of BC to have a stronger influence on LTS than the other climate drivers is most likely due to its strong absorption of solar radiation, which causes local heating of the atmosphere. Contrasting changes over land and ocean under global warming have been discussed for various climate change variables[27,28], and is often linked to the stronger surface heating over land than over ocean, and to the difference in availability of moisture.

Another factor that may contribute to changes in surface sensible heat is the near-surface winds. Figure 7 shows that also in this aspect BC differs from the other climate drivers investigated in PDRMIP. While the regional variations (Fig. 7c) are larger than for sensible heat and LTS changes, the $BC \times 10$ experiment causes an average reduction in surface wind over both land and ocean (Fig. 7a). For the other climate drivers, the mean surface wind changes are weak or similar in magnitude as the $CO_2 \times 2$ experiment (see Supplementary Fig. 9). In regions with the strongest changes in the surface winds the PDRMIP models to a large extent agree on the sign (Fig. 7b, c). At 1 km height the changes in zonal and meridional wind are generally stronger for BC than the other PDRMIP drivers (Supplementary Figs 10 and 11), but the difference is not as pronounced as for the near-surface wind changes. We conclude from the size and pattern of the changes in SH that these are mostly driven by changes in LTS, although some regional aspects of the SH change, especially in the BC experiments could be driven by changes in near-surface winds. A more detailed assessment of the drivers of SH change would require a thorough analysis of the different terms in the sensible heat parameterizations in the climate models. Such diagnostics are not available from the CMIP5 and PDRMIP experiments but would be worth more detailed study in the future.

**Importance of sensible heat and radiative cooling changes**. The present-day global-mean change in sensible heat relative to 1850 from the CMIP5 models can largely be explained by combining

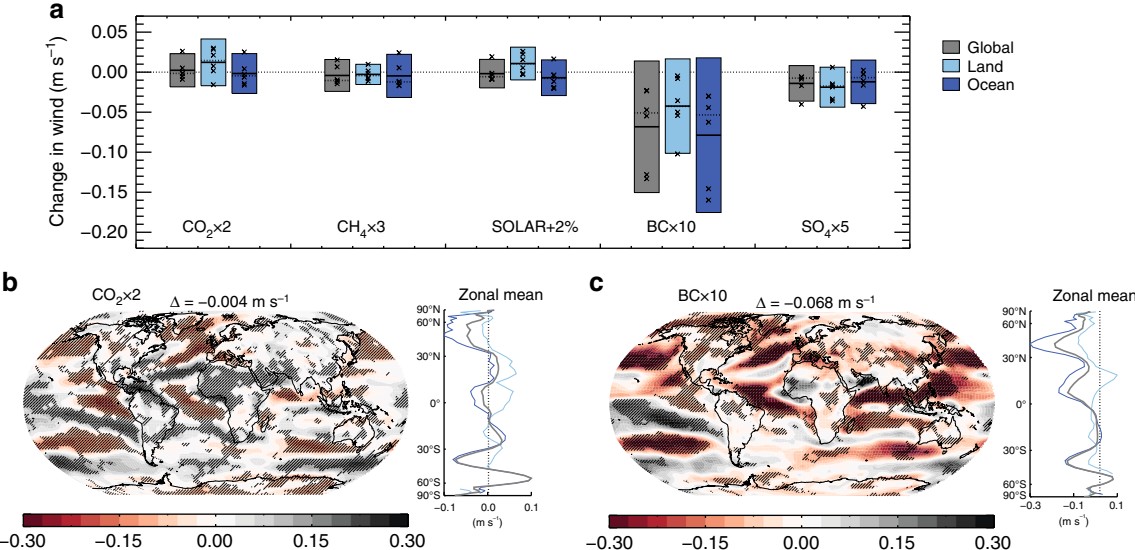

**Fig. 7** PDRMIP model-median changes in surface wind based on years 6–15 in fixed-SST simulations. **a** Shows global, land only and ocean only changes in surface wind, where individual crosses show individual model results, and the solid black line gives the model median. **b**, **c** show model median maps of surface wind changes for $CO_2 \times 2$ and $BC \times 10$, respectively, including zonal mean changes for the global (grey), land (light blue) and ocean (blue). Hatching is included where 75% or more of the models agree on the sign of the change

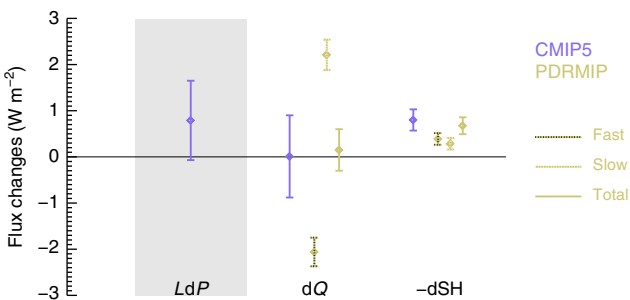

**Fig. 8** Historical changes in precipitation, sensible heat and radiative cooling from CMIP5 and PDRMIP simulations. The CMIP5 simulations are shown in purple, whereas fast and slow changes in sensible heat and radiative cooling derived from PDRMIP simulations are shown in light yellow. Uncertainties shown as one standard deviation. Sensible heat taken as change from atmosphere to surface ($-dSH$ in Eq. 1). Results from PDRMIP are shown for fast, slow and total changes. The uncertainties in the PDRMIP results for dSH and dQ include the range in the PDRMIP models, but not the temperature range among the CMIP5 models. Furthermore, the shown uncertainties in the combined fast and slow PDRMIP results are an upper bound since they are assumed to be independent

the observed surface temperature change, inducing a slow change in sensible heat, and the direct effect of changes in the abundance of the main drivers of climate change[29], inducing fast changes to sensible heat (see Methods section and Supplementary Table 2 for scaling the present-day drivers to the PDRMIP data and Fig. 8). At present, the multi-model mean contribution from fast changes to sensible heat is larger than the slow changes. For the fast changes, aerosols contribute about two-thirds of the total.

The implication of sensible heat changes balancing precipitation change is that atmospheric radiative cooling, at present, is a negligible term according to Eq. 1. This can be explained by atmospheric radiative heating occurring on a fast timescale (Fig. 1a), compensating for the radiative cooling as the surface and troposphere system warms under global warming[30,31] (and see Fig. 8). These two terms are individually about two times

larger in magnitude than the sensible heat and precipitation changes over the historical period (see Fig. 8 and Methods section for further details). The global-mean CMIP5 simulated precipitation change over the historical period has a much larger uncertainty than that expected in the future, due to the currently compensating fast radiative heating and slow radiative cooling terms; by contrast, in the future, the better constrained slow radiative cooling term with lower range among the models will dominate. The expected reduction in BC emissions[22] will further reduce the atmospheric radiative heating contribution to the uncertainty, since modelling of BC involves large uncertainties[32–36]. The sensible heat term has low uncertainty over the historical period due to the temperature driven portion and the fast changes from aerosols, each having a relatively small uncertainty and the same sign (see Fig. 8), whereas uncertainties in sensible heat increases with reductions in aerosol emissions.

## Discussion

Current and future precipitation changes can be understood in various ways: by radiative forcing agents[37], by fast and slow changes as shown in Fig. 8, or directly by terms given Eq. 1. Here we have used the latter approach to show that current precipitation changes since preindustrial time are dominated by sensible heat changes and the net radiative cooling is a negligible term. We emphasize that this is true only for current conditions; the magnitude of the individual fast and slow terms of the radiative cooling are larger than the sensible heat term and thus the radiative cooling terms generally dominate over the historical period. At other periods such as in the 1970s (see Fig. 2) the fast component of the radiative cooling dominates the response. This is because of a small temperature increase compared with the present (of the order of a quarter of current warming)[3], whereas changes in radiative forcing due to the drivers such as $CO_2$, $CH_4$ and BC have had a stronger relative increase (more than 50% of the present radiative forcing). By contrast, the future surface temperature increase towards 2100 will be three times that of the historical change, if we follow the RCP8.5 pathway, according to multi-model results from CMIP5. The fast component of the radiative cooling will strengthen much less than three times

the current value. In the RCP8.5 scenario the fast term will be enhanced by a strong increase in $CO_2$ and partly $CH_4$, but aerosols will be substantially reduced[38,39]. Currently aerosols enhance the fast component of the radiative cooling in the same direction as greenhouse gases and generally reduce the surface temperature. In the future, the opposite situation will occur, causing a much stronger increase in the slow component compared with the fast component of the radiative cooling. Furthermore, for all time periods, uncertainties in modelled precipitation change are dominated by the radiative cooling term since the magnitude of the fast and slow terms are large and of opposite magnitude. Over the historical period, the CMIP5 ensemble shows low variability in simulated changes to sensible heat. This is due to the driving terms, from temperature changes and the climate drivers, being comparable. For future simulations, with the expected reduction in the atmospheric aerosol abundance, the model diversity can be expected to increase.

## Methods

**PDRMIP data**. In this work, simulations from ten PDRMIP models are used (CanESM, GISS ModelE, HadGEM2, HadGEM3, IPSL-CM5, MPI-ESM, NCAR CESM1/CAM4, NCAR CESM1/CAM5, NorESM, SPRINTARS). Further details are described in Table 3 of the PDRMIP overview paper[19] and the methodology to calculate precipitation changes, atmospheric absorption, and radiative forcing uses fixed-SST and coupled climate simulations for fast, slow and total changes[20,40,41]. The length of the fixed-SST simulations is a minimum of 15 years and the fully coupled climate simulations are 100 years long. The fast response is derived from years 6–15 of the fixed-SST simulations and the total response is derived from the last 50 years of the coupled climate simulations. The difference between the response from the total and fast is used to derive the slow response. Atmospheric absorption (negative radiative cooling) is derived from the difference between top of the atmosphere and surface net radiative fluxes. Each model has performed one ensemble member simulations for the fixed-SST, as well as for the coupled climate simulations for each of the PDRMIP drivers and a reference simulation. Results from the five core PDRMIP perturbation simulations are used in this study; a doubling of $CO_2$ concentration (denoted $CO_2 \times 2$), tripling of $CH_4$ concentration ($CH_4 \times 3$), 2% increase in solar insolation (SOLAR + 2%), 10-fold increase in BC concentration or emissions (BC × 10), and 5-fold increase in $SO_4$ concentrations or emissions ($SO_4 \times 5$).

**Historical changes in sensible heat and radiative cooling**. To derive historical changes in radiative cooling and sensible heat based on the PDRMIP simulation in Fig. 8, the PDRMIP have been combined with historical changes of the climate drivers. The slow changes in sensible heat and atmospheric radiative cooling are derived from the PDRMIP mean results of slow changes per global-mean surface temperature changes multiplied by historical temperature change. For the period 2010–2016 relative to 1860–1866 a temperature change of 1.09 K has been calculated from the CMIP5 models, which have a slightly higher temperature change relative to observations[42]. The fast changes in sensible heat and atmospheric radiative cooling are derived from the PDRMIP mean results for the individual PDRMIP drivers and relative change in forcing compared to the PDRMIP perturbed forcing. We assume that the fast response of sensible heat and radiative cooling add linearly over the climate drivers for the historical period. For current forcing values relative to 1850 from IPCC AR5[29] are adopted. Supplementary Tables 1-4 summarizes values used to produce the results shown in Fig. 8. Supplementary Tables 1 and 2 are adopted to calculate the sensible heat changes in Fig. 8. Supplementary Table 1 provides the PDRMIP results and Supplementary Table 2 how historical drivers are combined with the PDRMIP results. Supplementary Tables 3 and 4 are adopted to calculate the radiative cooling changes in Fig. 8. Supplementary Table 3 provides the PDRMIP results and Supplementary Table 4 shows how historical drivers are combined with the PDRMIP results.

**CMIP5 data**. A selection of 24 models have been included from the CMIP5 data set[18]. A list of these models is given in the Supplementary Table 6. Results from the historical, RCP2.6 and RCP8.5 experiments have been used in this study.

**Data availability**. All PDRMIP model results used for the present study are available to the public through the Norwegian FEIDE data storage facility. For more information, see http://cicero.uio.no/en/PDRMIP. The CMIP5 data are available through the portal, the Earth System Grid-Center for Enabling Technologies (ESG-CET), on the page http://pcmdi9.llnl.gov/.

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

## Acknowledgements

G.M., B.H.S. and C.W.S were funded by the Research Council of Norway, through the grant NAPEX (229778). Supercomputer facilities were generously provided by NOTUR. Ø.H. acknowledges funding from the Research Council of Norway through the GREAT (GREenhouse gases, Aerosols and lower atmospheric Turbulence) project (grant no. 275589). D.S. thanks the NASA High-End Computing Program through the NASA Center for Climate Simulation at Goddard Space Flight Center for computational resources. M.K. and A.V. are supported by the Natural Environment Research Council under grant number NE/K500872/1. Simulations with HadGEM3-GA4 were performed using the MONSooN system, a collaborative facility supplied under the Joint Weather and Climate Research Programme, which is a strategic partnership between the Met Office and the Natural Environment Research Council. T.T. was supported by the supercomputer system of the National Institute for Environmental Studies, Japan, the Environment Research and Technology Development Fund (S-12-3) of the Ministry of the Environment, Japan and JSPS KAKENHI Grant Number JP15H01728 and JP15K12190. D.O. and A.K. were supported by the Norwegian Research Council through the projects EVA (grant no. 229771) and EarthClim (207711/E10), and NOTUR (nn2345k) and NorStore (ns2345k) projects. T.B.R. was supported by NERC training award NE/K007483/1, and acknowledges use of the MONSooN system. Computing resources for J.-F.L. (ark:/85065/d7wd3xhc) were provided by the Climate Simulation Laboratory at NCAR's Computational and Information Systems Laboratory, sponsored by the National Science Foundation and other agencies. Computing resources for the simulations with the MPI-ESM model were provided by the German Climate Computing Center (DKRZ), Hamburg. Computing resources for O.B. were provided by GENCI at the TGCC under allocation gen2201. T.A. was supported by the Newton Fund through the Met Office Climate Science for Service Partnership Brazil (CSSP Brazil). We acknowledge the World Climate Research Programme's Working Group on Coupled Modelling, which is responsible for CMIP, and we thank the climate modeling groups (listed in Supplementary Table 6 of this paper) for producing and making available their model output. For CMIP the U.S. Department of Energy's Program for Climate Model Diagnosis and Intercomparison provides coordinating support and led development of software infrastructure in partnership with the Global Organization for Earth System Science Portals.

## Author contributions

G.M. designed and performed most of the analysis. C.W.S. and Ø.H. performed the analysis on LTS and surface wind changes, respectively. T.A., O.B., G.F., D.F., M.K., V.K., A.K., J.-F.L., D.O., T.B.R., B.H.S, D.S., T.T. and A.V. contributed PDRMIP model data. G. M. led the writing of the manuscript with contributions from all authors, with particular contribution from T.A., O.B., P.M.F., B.H.S. and K.P.S. on framing the results.

## Additional information

**Competing interests:** The authors declare no competing interests.

