## [Peer Review File · Nature Communications]

Editorial Note: This manuscript has been previously reviewed at another journal that is not operating a transparent peer review scheme. This document only contains reviewer comments and rebuttal letters for versions considered at Nature Communications. Mentions of prior referee reports have been redacted.

Reviewers' comments:

Reviewer #1 (Remarks to the Author):

Since my previous review the authors have clarified some of the methods, have added a bit more physical interpretation of the results, and have added some extra discussion in places to put the results in perspective. Nonetheless, I still feel that the overall message of the paper lacks broader significance and that the paper is framed in a way that somewhat distracts from its truly new contributions. Those contributions include the analysis and discussion of how surface sensible heat flux changes over the historical period, from the perspective of various forcing agents, and how that relates to the global hydrologic cycle. Comparison of those changes with those in atmospheric radiative heating and under projected future scenarios is important, but in my opinion is given too much emphasis with regard to the main message of the paper.

Although the broader conclusions of the paper are not terribly surprising and arguably not that significant in a broader context, I feel that the paper's contributions in terms of analyzing sensible heat flux and the various forcing agents is of interest to specialists and therefore potentially suitable for publication in Nature Communications. On that note, I recommend that 1) the paper be reframed in terms of an analysis of sensible heat flux and the hydrologic cycle over the historical period, and 2) that even more physical interpretation of the sensible heat flux changes be given.

With regard to the first recommendation, I suggest not framing the paper as "sensible heat flux is the dominant contributor to precipitation changes." As the authors state in several places, the fast and slow radiative heating contributions are both large but cancel, allowing the magnitude of precipitation change at the end of the historical period to be consistent with that in sensible heat flux. While it is important to point this out, it does not suggest that sensible heat flux is a "driver" or "cause" of the precipitation change from a physical perspective (see also more detailed comments below). Essentially, the paper should be presented with a focus on the changes in sensible heat flux and alongside that a comparison with how radiative heating changes. Then, these changes should be compared with those under the future scenario to put the historical changes in perspective (as the paper already does). The title should be changed to something like "Changes in sensible heat and the global hydrological cycle over the historical period" and rest of the text modified accordingly to be consistent with that story.

With regard to the second recommendation, I appreciate the additional analysis on lower tropospheric stability, but feel that more justification and analysis/discussion can be given. First, I would think that the skin and near-surface air temperature difference (rather than 1000 and 780 hPa levels) would be more relevant for sensible heat flux. Are the results robust to the definition of the temperature difference? Further investigating the changes in surface and near-surface temperature separately would also be useful. What about the importance of other variables, such as wind speed, and the effect of rapid adjustments of, for example, clouds on the sensible heat flux change in response to the different types of forcing? Furthermore, additional maps like Fig. 5 b,c for all forcing scenarios (perhaps added in Supplementary) would be beneficial. It is also unclear if Fig. 5 is showing total or fast/slow changes and whether the results would be dependent on that.

Specific comments:

Abstract, L39-40: I disagree with this statement, as the reason for the smaller historical precipitation change is fundamentally less warming and the importance of different forcing agents (i.e., more aerosols). The characteristics of radiative cooling and sensible heat changes are just a consequence of those fundamental differences.

L75: "role of SH in causing current precipitation changes" - I think it is misleading to phrase it this way because it is not clear that a decrease in SH "causes" precipitation to increase. Rather, the SH decrease and precipitation increase are two adjustments of the atmospheric energy budget. Both changes are constrained by atmospheric energy balance, but it's not clear if one is directly causing the change in the other without further analysis.

L80: "in driving" - Similar comment as above. I think it is better to say "the changes in radiative cooling and sensible heat that are associated with historical and future precipitation changes."

L85-87: It is still unclear what "correlates" means here. If what is being discussed is a general physical relationship between precipitation change and absorption/forcing, rather than an actual computed statistical correlation, then I recommend "scales" or "is proportional to" instead.

L120: Similar to above comments with "driven by"

L125: Similar to above comments with "the cause of"

L130-131: I am uncertain of the value of Supp. Fig. 3a. In terms of understanding model spread, it may be more informative to correlate the black carbon burden directly with the sensible heat flux and radiative responses (total, slow, fast) instead, as BC spread is potentially more relevant for SH. Also, this statement appears with no context given as to why BC burden is investigated.

L163-165: These signs do not seem consistent with the definition of LTS described in Fig. 5 (1000 hPa minus 780 hPa). Also, the phrasing on L165 is awkward and should be revised.

L187-189: My previous comment about this statement has not been addressed: It's unclear to me how the sign of the fast and slow responses alone translate to the uncertainty of their sum. I would think that the degree of correlation or anticorrelation between the two as well as the magnitude of spread for each would be more relevant indications. It is not clear what was added to the figure caption to address this.

L194-195: I would think that "through an energy balance" is related to the first two ways.

L204-211: I appreciate the added discussion regarding the future scenario, but think that more similar discussion about the how the sensible heat flux may be different would be beneficial.

Typos/grammar:

L44: change "to other aerosols" to "compared to other aerosols"

L96: change "describes" to "describe"

Supp. Fig. 1: change "each models" to "each model"

Supp. Fig. 2: on plot change "RCP45" to "RCP4.5"

L152-153: change "southern America" to "South America"

Supp. Fig. 5: typo "of6sensible"

Table S4: Last sentence is confusing. Please reword.

L199: Do you mean future period rather than historical period?

Reviewer #2 (Remarks to the Author):

The revised manuscript by Myhre et al. has improved and is clearer than the initially submitted

version. In particular, the conclusions are now presented in a more balanced way and are no longer misleading. I am still disappointed by the rather superficial analysis of the mechanisms controlling the changes in sensible heat flux, which are at the core of the physical argument. The authors now present evidence for changes in lower tropospheric stability, which is rather obvious, however they fail to discuss why those are changing and why they are forcing dependent. One may argue that this is beyond the scope of this manuscript but I am not convinced it is, since the paper puts a specific focus on sensible heat as a key contributor to precipitation changes. Thus, understanding why it changes seems crucial. I am also not convinced that the metric chosen to illustrate changes in lower tropospheric stability is ideal. Please find my detailed comments below.

Physical mechanisms: I think the metric used to quantify LTS is not ideal since it is based on pressure levels. The warming in the lower troposphere induces an expansion of the 1000-780hPa thickness. In other words, the height difference between 780 and 1000hPa generally increases with warming and thus their temperature difference increases even in the absence of any change in the lapse rate $-dT/dz$. To disentangle the effects one would need to calculate $-dT/dz$ rather than dT/dp . As described above it would be essential to understand why the lapse rate changes the way it does and why it shows such a pronounced ocean-land contrast and forcing dependence. If an additional analysis is impossible due to model output limitations in PDRMIP, at least a detailed discussion of those ocean-land patterns based on the scientific literature would be helpful. The literature on the theoretical foundation of land-sea warming contrast should provide insights at least for the all-forcing historical simulations.

Model dependence: I think the authors still hide some model disagreement in the main text, which becomes only evident when looking at Fig.S1. In line 115 and 116 they make a clear statement for the multi-model mean (without specifying that it applies to the multi-model mean) and fail to acknowledge that the results in individual models shown in Fig.S1 are inconsistent with this statement. Some models show no cancellation of dQ up to now. I think it vital to emphasize these model discrepancies, like in the previous response to my comments and ideally even explain the reasons for those differences based on the framework used to produce Fig.6.

Supplementary material: The paragraph explaining Table S1-S4 is still insufficient to reproduce and understand the changes shown in Table S1-S4. This analysis is vital to connect the results based on CMIP5 to the ones based on PDRMIP. Thus, the results given in the tables should be discussed and each step carefully documented. For instance, I do not understand what the footnote to Table S2 means and how this scaling is exactly done.

Line 113-114: Is this reduction also mostly occurring over land in the transient runs?

Line 150: I don't exactly understand the second half of this sentence.

Line 166-167: Any idea why BC affects the LTS in that way and produces this pattern?

Line 168-169: Is this destabilization over land consistent with the literature? Please provide references.

Line 191-192: Why is the change less uncertain if the two have the same sign?

Fig.4: Is it expected that "CO2" and "Solar" have such a similar pattern. I would have thought they have different effects on the lapse rate.

Reviewer #1 (Remarks to the Author):

Since my previous review the authors have clarified some of the methods, have added a bit more physical interpretation of the results, and have added some extra discussion in places to put the results in perspective. Nonetheless, I still feel that the overall message of the paper lacks broader significance and that the paper is framed in a way that somewhat distracts from its truly new contributions. Those contributions include the analysis and discussion of how surface sensible heat flux changes over the historical period, from the perspective of various forcing agents, and how that relates to the global hydrologic cycle. Comparison of those changes with those in atmospheric radiative heating and under projected future scenarios is important, but in my opinion is given too much emphasis with regard to the main message of the paper.

Although the broader conclusions of the paper are not terribly surprising and arguably not that significant in a broader context, I feel that the paper's contributions in terms of analyzing sensible heat flux and the various forcing agents is of interest to specialists and therefore potentially suitable for publication in Nature Communications. On that note, I recommend that 1) the paper be reframed in terms of an analysis of sensible heat flux and the hydrologic cycle over the historical period, and 2) that even more physical interpretation of the sensible heat flux changes be given.

With regard to the first recommendation, I suggest not framing the paper as "sensible heat flux is the dominant contributor to precipitation changes." As the authors state in several places, the fast and slow radiative heating contributions are both large but cancel, allowing the magnitude of precipitation change at the end of the historical period to be consistent with that in sensible heat flux. While it is important to point this out, it does not suggest that sensible heat flux is a "driver" or "cause" of the precipitation change from a physical perspective (see also more detailed comments below). Essentially, the paper should be presented with a focus on the changes in sensible heat flux and alongside that a comparison with how radiative heating changes. Then, these changes should be compared with those under the future scenario to put the historical changes in perspective (as the paper already does). The title should be changed to something like "Changes in sensible heat and the global hydrological cycle over the historical period" and rest of the text modified accordingly to be consistent with that story.

With regard to the second recommendation, I appreciate the additional analysis on lower tropospheric stability, but feel that more justification and analysis/discussion can be given. First, I would think that the skin and near-surface air temperature difference (rather than 1000 and 780 hPa levels) would be more relevant for sensible heat flux. Are the results robust to the definition of the temperature difference? Further investigating the changes in surface and near-surface temperature separately would also be useful. What about the importance of other variables, such as wind speed, and the effect of rapid adjustments of, for example, clouds on the sensible heat flux change in response to the different types of forcing? Furthermore, additional maps like Fig. 5 b,c for all forcing scenarios (perhaps added in Supplementary) would be beneficial. It is also unclear if Fig. 5 is showing total or fast/slow changes and whether the results would be dependent on that.

Response: We appreciate the Reviewers perspective of the manuscript and suggestions for improvements. The main changes to the manuscript have been directed to historical sensible heat changes, with further analysis of the CMIP5 simulations but also additional work on the PDRMIP data to interpret the sensible heat changes. We have made the following major changes to the manuscript.

*The title is changed. We have made the title slightly shorter than the suggestion: '*Sensible heat and the global hydrological cycle*'

*The robustness of the lower tropospheric stability (LTS) with regard to layers has been investigated. Supplementary figures have been added and show that LTS behaves in a relatively similar way for different choices of vertical layers.

*A new figure of surface wind is included with associated text to investigate causes of changes in sensible heat.

*A new figure on sensible heat changes in the CMIP5 models over the historical period. We link this figure to sensible heat changes from the various PDRMIP drivers.

*Supplementary figures of wind at 1 km and LTS are added. The new supplementary figure on LTS is given for all PDRMIP drivers as suggested by the Reviewer in addition to investigations to the sensitivity of choice of levels in the lower atmosphere.

Specific comments:

Abstract, L39-40: I disagree with this statement, as the reason for the smaller historical precipitation change is fundamentally less warming and the importance of different forcing agents (i.e., more aerosols). The characteristics of radiative cooling and sensible heat changes are just a consequence of those fundamental differences.

Response: We disagree with the Reviewer on this point. The RCP2.6 scenario has about or slightly stronger anthropogenic warming in the future relative to the present day, but future precipitation changes are much stronger, see Figure 2a.

L75: "role of SH in causing current precipitation changes" - I think it is misleading to phrase it this way because it is not clear that a decrease in SH "causes" precipitation to increase. Rather, the SH decrease and precipitation increase are two adjustments of the atmospheric energy budget. Both changes are constrained by atmospheric energy balance, but it's not clear if one is directly causing the change in the other without further analysis.

Response: We agree that 'causing' may be a too strong wording and have changed this to 'contributing to'.

L80: "in driving" – Similar comment as above. I think it is better to say "the changes in radiative cooling and sensible heat that are associated with historical and future precipitation changes."

Response: Text changed as suggested.

L85-87: It is still unclear what "correlates" means here. If what is being discussed is a general physical relationship between precipitation change and absorption/forcing, rather than an actual computed statistical correlation, then I recommend "scales" or "is proportional to" instead.

Response: We have replaced 'correlates' with 'scales' as suggested.

L120: Similar to above comments with "driven by"

Response: The sentence have been rewritten to '*For future precipitation changes, on the other hand, the contribution from enhanced radiative cooling dominates, with sensible heat contributing only modestly in the RCP8.5 scenario*'

L125: Similar to above comments with "the cause of"

Response: 'the cause of' is replaced by 'has a large contribution to'

L130-131: I am uncertain of the value of Supp. Fig. 3a. In terms of understanding model spread, it may be more informative to correlate the black carbon burden directly with the sensible heat flux and radiative responses (total, slow, fast) instead, as BC spread is potentially more relevant for SH. Also, this statement appears with no context given as to why BC burden is investigated.

Response: Supp Fig 3a was added based on a request from Reviewer 2 to add more investigations on the CMIP5 simulations of the historical precipitation changes. We have a similar figure on BC burden and changes in sensible heat (see below), without providing any further understanding of the historical changes in the CMIP5 simulations in our view. We have therefore not included the figure in the supplementary. Separation of total, slow and fast is not available for CMIP5 runs and is the big advantage with the PDRMIP simulations. The split into total, slow and fast for the PDRMIP simulations is already highlighted in the manuscript. We have added the following sentence to describe the background of adding Supp Fig 3. 'While BC has been shown to affect precipitation in substantially different ways than other drivers of climate change^{4,6,23}, it is difficult to discern any impact on simulated historical precipitation change in CMIP5 due to differences in BC abundances (Figure S3).'

L163-165: These signs do not seem consistent with the definition of LTS described in Fig. 5 (1000 hPa minus 780 hPa). Also, the phrasing on L165 is awkward and should be revised.

Response: We agree that the sentence on L165 could be improved and this is rewritten to the following: 'When the LTS is negative, this reflects an increase in the lower atmospheric stability.' We have ensured that the sign of LTS is consistent in the text, figure and caption.

L187-189: My previous comment about this statement has not been addressed: It's unclear to me how the sign of the fast and slow responses alone translate to the uncertainty of their sum. I would think that the degree of correlation or anticorrelation between the two as well as the magnitude of spread for each would be more relevant indications. It is not clear what was added to the figure caption to address this.

Response: In the revised manuscript, we added the following to the caption of the new Fig 8: 'Furthermore, the shown uncertainties in the combined fast and slow PDRMIP results are an upper bound since they are assumed to be independent.'

L194-195: I would think that "through an energy balance" is related to the first two ways.

Response: We rephrased the sentence by 'or directly by terms given in Equation 1'

L204-211: I appreciate the added discussion regarding the future scenario, but think that more similar discussion about the how the sensible heat flux may be different would be beneficial.

Response: The following sentence has been added: *'Over the historical period, the CMIP5 ensemble shows low variability in simulated changes to sensible heat. This is due to the driving terms, from temperature changes and the climate drivers, being comparable. For future simulations, with the expected reduction in the atmospheric aerosol abundance, the model diversity can be expected to increase.'*

Typos/grammar:

L44: change "to other aerosols" to "compared to other aerosols"

Response: Typos/grammar corrected.

L96: change "describes" to "describe"

Response: Typos/grammar corrected.

Supp. Fig. 1: change "each models" to "each model"

Response: Typos/grammar corrected.

Supp. Fig. 2: on plot change "RCP45" to "RCP4.5"

L152-153: change "southern America" to "South America"

Response: Typos/grammar corrected.

Supp. Fig. 5: typo "of6sensible"

Response: Typos/grammar corrected.

Table S4: Last sentence is confusing. Please reword.

Response: Typos/grammar corrected.

L199: Do you mean future period rather than historical period?

Response: Changed to 'Current and future'

Reviewer #2 (Remarks to the Author):

The revised manuscript by Myhre et al. has improved and is clearer than the initially submitted version. In particular, the conclusions are now presented in a more balanced way and are no longer misleading. I am still disappointed by the rather superficial analysis of the mechanisms controlling the changes in sensible heat flux, which are at the core of the physical argument. The authors now present evidence for changes in lower tropospheric stability, which is rather obvious, however they fail to discuss why those are changing and why they are forcing dependent. One may argue that this is beyond the scope of this manuscript but I am not convinced it is, since the paper puts a specific focus on sensible heat as a key contributor to precipitation changes. Thus, understanding why it changes seems crucial. I am also not convinced that the metric chosen to illustrate changes in lower tropospheric stability is ideal. Please find my detailed comments below.

Response: We appreciate the Reviewers view on the improvements of the manuscript and appreciate the suggestions for further improvements. We have made the following major changes to the manuscript:

*We have added more physical interpretations of the results and further factors that may influence the sensible heat. This includes a new figure on wind changes in the main part of the manuscript and several figures in the supplementary.

*We have further investigated the choice of the LTS metric and found that the figure presented in the earlier revised manuscript was robust with regard to comments by the two reviewers.

*We have added a new figure on the geographical distribution of the change in sensible heat from the CMIP5 models into the main part of the manuscript.

Physical mechanisms: I think the metric used to quantify LTS is not ideal since it is based on pressure levels. The warming in the lower troposphere induces an expansion of the 1000-780hPa thickness. In other words, the height difference between 780 and 1000hPa generally increases with warming and thus their temperature difference increases even in the absence of any change in the lapse rate $-dT/dz$. To disentangle the effects one would need to calculate $-dT/dz$ rather than dT/dp .

Response: We agree to the fact that the thickness of the 1000-780hPa layer will expand under warming. We believe that this will only constitute a minor influence on the LTS calculation, and we have investigated this in PDRMIP model standard output. As an example, we have looked at LTS changes for one of the models (CAM4) for BASE, CO2x2 and BCx10. We find that the global mean geopotential thickness between the two pressure levels are 0.72 and 0.34 meters higher for CO2x2 and BCx10, respectively than for BASE. Giving LTS changes in % for more easy comparison of dT/dp and dT/dz we then find that for CO2x2, the LTS changes by the two definitions are $dT/dp = -0.57\%$ and $dT/dz = -0.67\%$. For BCx10, they are $dT/dp = -1.28\%$ and $dT/dz = -1.33\%$. In other words, while there are indeed differences between the two ways to calculate LTS we find that difference between dT/dp and dT/dz small compared to the major differences as illustrated in Figure 5.

As described above it would be essential to understand why the lapse rate changes the way it does and why it shows such a pronounced ocean-land contrast and forcing dependence. If an additional analysis is impossible due to model output limitations in PDRMIP, at least a detailed discussion of those ocean-land patterns based on the scientific literature would be helpful. The literature on the theoretical foundation of land-sea warming contrast should provide insights at least for the all-forcing historical simulations.

Response: We have added the following text on the lapse-rate part:

‘Contrasting changes over land and ocean under global warming have been discussed for various climate change variables^{27,28}, and is often linked to the stronger surface heating over land than over ocean, and to the difference in availability of moisture. The tendency of BC to have a stronger influence on LTS than the other climate drivers is likely due to its strong absorption of solar radiation, which causes local heating of the atmosphere.’

Model dependence: I think the authors still hide some model disagreement in the main text, which becomes only evident when looking at Fig.S1. In line 115 and 116 they make a clear statement for the multi-model mean (without specifying that it applies to the multi-model mean) and fail to acknowledge that the results in individual models shown in Fig.S1 are inconsistent with this statement. Some models show no cancellation of dQ up to now. I think it vital to emphasize these model discrepancies, like in the previous response to my comments and ideally even explain the reasons for those differences based on the framework used to produce Fig.6.

Response: It was not our intention to hide anything. We have made it clear that the statement is for multi-model mean and have explicitly added the following sentence to draw the attention to the individual models:

‘Note that the multi-model mean of increase in precipitation is equal to the reduction in surface sensible heat for the period from 2010 to 2016; however, this is not the case for all individual CMIP5 models (see Supplementary Figure 1).’

Supplementary material: The paragraph explaining Table S1-S4 is still insufficient to reproduce and understand the changes shown in Table S1-S4. This analysis is vital to connect the results based on CMIP5 to the ones based on PDRMIP. Thus, the results given in the tables should be discussed and each step carefully documented. For instance, I do not understand what the footnote to Table S2 means and how this scaling is exactly done.

Response: We have added some sentences how the Table S1-S4 are adopted. For Table S1-S2 the following sentences have been added and similar changes have been made for Table S3-S4.

‘Tables S1 and S2 are adopted to calculate the sensible heat changes in Figure 7. Table S1 provides the PDRMIP results and Table S2 how historical drivers are combined with the PDRMIP results’.

Line 113-114: Is this reduction also mostly occurring over land in the transient runs?

Response: Based on this comment we have added a new figure into the main manuscript (Figure 3). The figure is discussed in the text and the PDRMIP simulated changes (shown now in Figure 5) resembles many of the historical changes simulated in the CMIP5 models.

Line 150: I don't exactly understand the second half of this sentence.

Response: The sentence has been corrected to: *‘The reduction over ocean dominates for the slow changes, whereas the changes over land and ocean almost cancel each other in the ensemble-mean for the fast changes²⁴.’*

Line 166-167: Any idea why BC affects the LTS in that way and produces this pattern?

Response: Based on a similar comment from Reviewer 1 we have added in the paragraph the following: *'The tendency of BC to have a stronger influence on LTS than the other climate drivers is likely due to its strong absorption of solar radiation, which causes local heating of the atmosphere.'*

Line 168-169 Is this destabilization over land consistent with the literature? Please provide references.

Response: Based on a similar comment from Reviewer 1 we have added in the paragraph the following: *'Contrasting changes over land and ocean under global warming have been discussed for various climate change variables^{27,28}, and is often linked to the stronger surface heating over land than over ocean, and to the difference in availability of moisture.'*

Line 191-192: Why is the change less uncertain if the two have the same sign?

Response: We have added that both terms also have small relative uncertainty.

Fig.4 Is it expected that "CO2" and "Solar" have such a similar pattern. I would have thought they have different effects on the lapse rate.

Response: We agree that the slow is quite similar, but the fast is much more dissimilar. In the revised version of the supplementary we added Figure 7, which shows the LTS for all PDRMIP drivers (and not only CO2 and BC such as in Figure 6).

REVIEWERS' COMMENTS:

Reviewer #1 (Remarks to the Author):

The authors have sufficiently addressed the main concerns from my previous review. I only have one remaining minor comment related to content (given below), as well as a list of minor word change suggestions/typos.

With regard to the LTS and wind analysis: The current analysis only shows changes in LTS and speculates that those changes are largely due to heating of the atmosphere. I am curious about the change in land surface/near-surface air temperature with BC forcing and whether that contributes to the change in LTS. Furthermore, I wonder if surface temperature changes can be linked to the reductions in wind (perhaps through a change in the land-sea temperature contrast) shown in Fig. 7. More generally, the potential mechanisms for changes in wind are not discussed and I imagine are quite complex. Perhaps a bit more can be said about this in the manuscript.

L86: Change "correlates" to "scales" if this statement has the same meaning as L84.

L126: Change "for predicted" to "for the predicted"

L130: Change "factor 3-4" to "factor of 3-4"

Fig. 4 caption, last line: "dP" should be "LdP" if energy units are given.

L178: "100 hPa" should be "1000 hPa"

Supp. Fig. 6 caption, line 5: Change "dome" to "some"

L179: Change "When the LTS is negative" to "When the LTS change is negative"

Reviewer #2 (Remarks to the Author):

The authors have satisfactorily addressed all my previous review comments and carefully tested the sensitivity to the definition of the lower tropospheric stability. The authors' interpretation on the role of surface winds, a newly added analysis, is not entirely clear to me. Based on the figures shown, it seems to be a negligible effect. I wonder whether the patterns in Fig.7 primarily show noise or a real forced response. The authors should provide a measure of robustness and add stippling to Fig.7 to illustrate whether this is a robust forced response pattern or only variability. Apart from this important minor point, I recommend accepting the revised manuscript for publication.

Reviewer #1 (Remarks to the Author):

The authors have sufficiently addressed the main concerns from my previous review. I only have one remaining minor comment related to content (given below), as well as a list of minor word change suggestions/typos.

With regard to the LTS and wind analysis: The current analysis only shows changes in LTS and speculates that those changes are largely due to heating of the atmosphere. I am curious about the change in land surface/near-surface air temperature with BC forcing and whether that contributes to the change in LTS. Furthermore, I wonder if surface temperature changes can be linked to the reductions in wind (perhaps through a change in the land-sea temperature contrast) shown in Fig. 7. More generally, the potential mechanisms for changes in wind are not discussed and I imagine are quite complex. Perhaps a bit more can be said about this in the manuscript.

Response: We have added a new figure in the Supplementary (Supplementary Figure 8) to show the surface temperature changes in two of the experiments along with temperature changes at 780 hPa. The following text has been added *‘Supplementary Figure 8 shows the surface (1000 hPa) and 780 hPa temperature changes in the CO₂x2 and BCx10 fixed-SST experiments. The land surface temperature changes at mid to high northern latitudes are somewhat larger in the CO₂x2 experiment than the BCx10 experiment, but at 780 hPa over regions with high BC abundance the temperature changes are larger in the BCx10 than the CO₂x2 experiment, explaining the LTS change patterns.’*

From Supplementary Figure 8 we do not find any evidence the land-ocean contrast in surface temperature change is a cause for the difference in surface wind changes between the CO₂x2 and BCx10 experiments.

As a response to Reviewer 2 we have added hatching to Figure 7 and some description in the text of the robustness of the results.

L86: Change “correlates” to “scales” if this statement has the same meaning as L84.

Response: Corrected

L126: Change “for predicted” to “for the predicted”

Response: Corrected

L130: Change “factor 3-4” to “factor of 3-4”

Response: Corrected

Fig. 4 caption, last line: “dP” should be “LdP” if energy units are given.

Response: Corrected

L178: “100 hPa” should be “1000 hPa”

Response: Corrected

Supp. Fig. 6 caption, line 5: Change “dome” to “some”

Response: Corrected

L179: Change “When the LTS is negative” to “When the LTS change is negative”

Response: Corrected

Reviewer #2 (Remarks to the Author):

The authors have satisfactorily addressed all my previous review comments and carefully tested the sensitivity to the definition of the lower tropospheric stability. The authors’ interpretation on the role of surface winds, a newly added analysis, is not entirely clear to me. Based on the figures shown, it seems to be a negligible effect. I wonder whether the patterns in Fig.7 primarily show noise or a real forced response. The authors should provide a measure of robustness and add stippling to Fig.7 to illustrate whether this is a robust forced response pattern or only variability. Apart from this important minor point, I recommend accepting the revised manuscript for publication.

Response: We have added hatching in Fig 7b and 7c on the geographical pattern of surface wind changes. As a result of this inclusion we have added the following sentence in the manuscript: ‘In regions with the strongest changes in the surface winds the PDRMIP models to a large extent agree on the sign (Fig 7b and 7c).’ In the caption the following have been added: ‘Hatching is included where 75 % or more of the models agree on the sign of the change.’